# Machine learning for prediction of postoperative nausea and vomiting in patients with intravenous patient-controlled analgesia

**Jae-Geum Shim[1,2], Kyoung-Ho Ryu[2], Eun-Ah Cho[2], Jin Hee Ahn[2], Yun Byeong Cha[2], Goeun Lim[2], Sung Hyun Lee[2]***

**1** Department of Anesthesiology and Pain Medicine, College of Medicine, Graduate School, Kyung Hee University, Seoul, Korea, **2** Department of Anesthesiology and Pain Medicine, Kangbuk Samsung Hospital, Sungkyunkwan University School of Medicine, Seoul, Korea

* hoho4321.lee@daum.net

## Abstract

### Background

Postoperative nausea and vomiting (PONV) is a still highly relevant problem and is known to be a distressing side effect in patients. The aim of this study was to develop a machine learning model to predict PONV up to 24 h with fentanyl-based intravenous patient-controlled analgesia (IV-PCA).

### Methods

From July 2019 and July 2020, data from 2,149 patients who received fentanyl-based IV-PCA for analgesia after non-cardiac surgery under general anesthesia were applied to develop predictive models. The rates of PONV at 1 day after surgery were measured according to patient characteristics as well as anesthetic, surgical, or PCA-related factors. All statistical analyses and computations were performed using the R software.

### Results

A total of 2,149 patients were enrolled in this study, 337 of whom (15.7%) experienced PONV. After applying the machine-learning algorithm and Apfel model to the test dataset to predict PONV, we found that the area under the receiver operating characteristic curve using logistic regression was 0.576 (95% confidence interval [CI], 0.520–0.633), k-nearest neighbor was 0.597 (95% CI, 0.537–0.656), decision tree was 0.561 (95% CI, 0.498–0.625), random forest was 0.610 (95% CI, 0.552–0.668), gradient boosting machine was 0.580 (95% CI, 0.520–0.639), support vector machine was 0.649 (95% CI, 0.592–0.707), artificial neural network was 0.686 (95% CI, 0.630–0.742), and Apfel model was 0.643 (95% CI, 0.596–0.690).

**Data Availability Statement:** De-identified and pre-processed data were deposited to a public data repository (https://github.com/jgshim/IV-PCA). All

data underlying the findings described in our paper are fully available without restriction.

**Funding:** The author(s) received no specific funding for this work.

**Competing interests:** The authors have declared that no competing interests exist.

**Abbreviations:** ANN, Artificial neural network; AUROC, Area under the receiver operating characteristic curve; CI, Confidence interval; DT, Decision tree; GBM, Gradient boosting machine; KNN, K-nearest neighbors; PCA, Patient-controlled analgesia; PONV, Postoperative nausea and vomiting; RF, Random forest; SVM, Support vector machine; TIVA, Total intravenous anesthesia.

## Conclusions

We developed and validated machine learning models for predicting PONV in the first 24 h. The machine learning model showed better performance than the Apfel model in predicting PONV.

## Background

Postoperative nausea and vomiting (PONV) is a common condition and is known to be a distressing side effect in patients [1]. The incidence of PONV is 30% and can be as high as 80% in high-risk patients [2, 3]. Although the mechanism of PONV is not clear, the use of perioperative opioids is known to be associated with it [4]. Nonetheless, opioid-based intravenous patient-controlled analgesia (IV-PCA) currently plays an important role in routine postoperative analgesic therapy [5–7]. Therefore, by accurately predicting PONV, patients can be warned of the risk of developing PONV, and clinicians can be assisted in making decisions about preventive treatment.

Apfel's risk score is a simple assessment tool derived to predict the 24-h rates of PONV [8, 9]. However, the Apfel model does not guarantee the accurate prediction of the risk of PONV, with limited discrimination and calibration properties [10, 11]. Recently, studies have used dynamic predictive models or machine learning to improve the predictive performance of PONV [12–15].

Machine learning is the application of artificial intelligence, whereby a computer algorithm automatically learns and improves from prior experience [16]. The machine learning algorithm produces an inferred function that can be used as the predictor of new data after sufficient training with known input and output values [17]. It may be used for prediction in the medical field. Recently, machine learning algorithms have shown high performance in various fields of medicine, such as diagnosis, prognosis, and clinical decision support [18–21].

To the best of our knowledge, no previous study has compared the performance of Apfel and machine learning methods in predicting PONV 24 h after surgery. We expect our research results to improve the prediction of PONV and quality of patient care.

## Methods and methods

### Study population

We collected data from patients (>19 years) after non-cardiac surgery under general anesthesia who received fentanyl-based IV-PCA at Kangbuk Samsung Hospital between July 2019 and July 2020. The exclusion criteria for this study were refusal to receive PCA and admission to the intensive care unit. This study was reviewed and approved by the Institutional Review Board (IRB No. 2020-08-001) of Kangbuk Samsung Hospital (Seoul, Korea). This study was conducted in accordance with the principles of the Declaration of Helsinki of the World Medical Association. The need for written informed consent was waived as this was a retrospective study of electronic medical records.

### Data collection

The rates of PONV at 1 day after surgery were measured with information on postoperative pain scores and other complications by the PCA team in our hospital. We also included patient characteristics as well as anesthetic, surgical, or PCA-related factors in the predictive models.

The continuous variables were age, body mass index (BMI), duration of anesthesia, and dosage of fentanyl in IV-PCA. The categorical variables were sex, history of motion sickness or PONV, American Society of Anesthesiologists (ASA) physical status, diabetes mellitus, hypertension, premedication, use of preintubation opioids, anesthetic agents (sevoflurane, desflurane, or TIVA), intraoperative remifentanil infusion, the use of intraoperative opioids (fentanyl or meperidine), emergency operation, laparoscopic surgery, type of surgery, adjuvant nefopam, and antiemetic (ramosetron) in IV-PCA. Continuous variables were transformed to values between 0 and 1 by minimum-maximum normalization, implemented in the caret package in the R software.

## Feature selection

Feature selection is the process of selecting features that contribute the most to our prediction variable, leading to improved performance. In this process, recursive feature elimination was used as a method that fits the random forest function in the core of the model and removes the weakest feature until the specified number of features is reached. Features are ranked by the model's feature importance by iteratively eliminating a small number of features per loop. To enable the machine learning algorithms to run efficiently, we only used the data features resulting from recursive feature elimination to train our machine learning models.

## Model assessment

To determine the goodness of the prediction ability, model performance was evaluated by comparing machine learning approaches to the Apfel model in terms of the area under the receiver operating characteristic curve (AUROC). The AUROC was plotted using the test dataset to understand the tradeoff in performance for different threshold values in imbalanced classification problems. We also compared the accuracy, sensitivity, and specificity.

The confusion matrix is used for summarizing the performance of a classification problem as shown in Table 1. Accuracy, sensitivity, and specificity are described in terms of true positive (TP), true negative (TN), false negative (FN) and false positive (FP).

The accuracy of model is the ratio of correct predictions to total predictions made and is defined as

Accuracy = (TN + TP) / (TN+TP+FN+FP)

The sensitivity of model is the proportion of actual positive cases that are correctly identified and is defined as

Sensitivity = TP / (TP + FN)

The specificity of model is the proportion of actual negative cases that are correctly identified and is defined as

Specificity = TN / (TN + FP)

**Table 1. Confusion matrix.**

|  |  | Actual | |
|---|---|---|---|
|  |  | Positive | Negative |
| Predicted | Positive | TP | TN |
|  | Negative | FP | FN |

TP, true positive; TN, true negative; FN, false negative; FP, false positive

## Statistical analysis

All statistical analyses and computations were performed using the R software version 3.6.3 (R Development Core Team, Vienna, Austria). The machine learning algorithm was implemented using the following packages: Caret (https://CRAN.R-project.org/package=caret), Xgboost (https://CRAN.R-project.org/package=xgboost), and Keras (https://CRAN.R-project.org/package=keras). The entire code of our study (https://github.com/jgshim/PONV) is provided.

Before applying the machine learning models, our data set was randomly divided into 70/30 training and test sets, as we did not want our models to overfit and generalize well. Specifically, 70% of the data was used for training prediction models, and 30% was used as the testing set for verification. A 10-fold cross-validation repeated three times was used to assess how the predictive model generalizes to an independent dataset. The missing data were imputed using the nearest neighbor imputation algorithms, where each missing value is replaced by a value obtained from related cases in the entire data set [22]. The synthetic minority oversampling technique method, addressing imbalanced classification problems, was used to oversample the minority class and balance the low incidence of PONV in the training set [23].

## Results

### Patient's characteristics

The sample group included 2,680 patients who received fentanyl-based IV-PCA for analgesia after non-cardiac surgery under general anesthesia at Kangbuk Samsung Hospital between July 2019 and July 2020. A total of 23 patients aged ≤18 years were excluded. In addition, 508 patients were excluded because they were subjected to regional anesthesia. As a result, a total of 2,149 patients satisfying all inclusion criteria were enrolled in the study. During the 24 h after surgery, 337 patients (15.7%) experienced PONV. The patient characteristics as well as anesthetic, surgical, or PCA-related variables are summarized in Table 2. The correlation analysis showed a weak positive correlation between motion sickness, laparoscopy, desflurane, and gynecology surgery and PONV, as shown in Fig 1. However, male sex and smoking status showed a weak negative correlation with PONV.

### Feature selection

We identified 21 variables, including patient characteristics as well as anesthetic, surgical, or PCA-related factors, from previous studies conducted to identify features that may contribute to PONV. Among these variables, anesthetics and type of surgery were categorical variables with more than two levels. As an input for our models, categorical variables with n levels were transformed into n variables, each with two levels. As a result, 34 variables were initially considered as input variables for the model.

The recursive feature elimination algorithm resulted in the final 13 factors contributing to PONV. Fig 2 shows the process of feature selection after the step of recursive feature elimination. On final feature selection, only 13 features were used as input variables in training the machine learning models for predicting PONV.

### Model performance

The predictive performance of various machine learning and Apfel models is shown in Table 3. After applying the test dataset for all machine learning techniques and the Apfel score to predict PONV, we found that the AUROC using logistic regression was 0.576 (95% confidence interval [CI], 0.520–0.633), k-nearest neighbor was 0.597 (95% CI, 0.537–0.656),

**Table 2. Dataset characteristics.**

| | All cases | No PONV | PONV | P-value |
|---|---|---|---|---|
| | (N = 2,149) | (n = 1,812) | (n = 337) | |
| **Patient characteristics** | | | | |
| Age (y) | 60 (47–71) | 61 (48–71) | 57 (43–68) | < 0.001 |
| Sex (female) | 1,248 (58.1%) | 997 (55.0%) | 251 (74.5%) | < 0.001 |
| BMI (kg/m²) | 24.3 (22.2–26.7) | 24.3 (22.2–26.7) | 24.2 (22.1–26.8) | 0.44 |
| History of smoking | 523 (24.3%) | 477 (26.3%) | 46 (13.6%) | < 0.001 |
| History of motion sickness | 133 (5.0%) | 64 (3.5%) | 49 (14.5%) | < 0.001 |
| History of PONV | 16 (0.7%) | 8 (0.4%) | 8 (2.4%) | < 0.001 |
| ASA physical status | | | | < 0.001 |
| ASA I | 504 (23.5%) | 401 (22.1%) | 103 (30.6%) | |
| ASA II | 1,013 (47.1%) | 847 (46.7%) | 166 (49.3%) | |
| ASA III | 632 (29.4%) | 564 (31.1%) | 68 (20.2%) | |
| Diabetes mellitus | 421 (19.6%) | 373 (20.6%) | 48 (14.2%) | 0.009 |
| Hypertension | 794 (36.9%) | 690 (38.1%) | 104 (30.9%) | 0.01 |
| **Anesthetic factor** | | | | |
| Duration of anesthesia (min) | 160 (115–220) | 165 (115–225) | 145 (105–195) | < 0.001 |
| Premedication (anticholinergic) | 1,809 (84.2%) | 1,510 (83.3%) | 299 (88.7%) | 0.02 |
| Preintubation opioid | 254 (11.8%) | 197 (10.9%) | 57 (16.9%) | 0.002 |
| Anesthetics (%) in G/A | | | | < 0.001 |
| Sevoflurane | 1,547 (72.0%) | 1,328 (73.3%) | 219 (65.0%) | |
| Desflurane | 479 (22.3%) | 369 (20.4%) | 110 (32.6%) | |
| TIVA | 123 (5.7%) | 115 (6.3%) | 8 (2.4%) | |
| Intraoperative remifentanil infusion (%) | 1,879 (87.4%) | 1,597 (88.1%) | 282 (83.7%) | 0.03 |
| Intraoperative opioids (fentanyl, meperidine) | 1,788 (83.2%) | 1,524 (84.1%) | 264 (78.3%) | 0.01 |
| **Surgical factor** | | | | |
| Emergency (%) | 199 (9.3%) | 180 (9.9%) | 19 (5.6%) | 0.02 |
| Laparoscopic surgery | 465 (21.6%) | 352 (19.4%) | 113 (33.5%) | < 0.001 |
| Type of surgery | | | | < 0.001 |
| Abdominal | 420 (19.5%) | 376 (20.8%) | 44 (13.1%) | |
| Thoracic | 213 (9.9%) | 181 (10.0%) | 32 (9.5%) | |
| Obstetric | 20 (0.9%) | 16 (0.9%) | 4 (1.2%) | |
| Gynecological | 401 (18.7%) | 295 (16.3%) | 106 (31.5%) | |
| Urology | 174 (8.1%) | 154 (8.5%) | 20 (5.9%) | |
| Brain | 10 (0.5%) | 10 (0.6%) | 0 (0.0%) | |
| Spine | 262 (12.2%) | 237 (13.1%) | 25 (7.4%) | |
| Shoulder | 111 (5.2%) | 80 (4.4%) | 31 (9.2%) | |
| Hip | 44 (2.0%) | 39 (2.2%) | 5 (1.5%) | |
| Upper and lower extremities | 444 (20.7%) | 382 (21.1%) | 62 (18.4%) | |
| Skin, soft tissue | 20 (0.9%) | 17 (0.9%) | 3 (0.9%) | |
| Others | 30 (1.4%) | 25 (1.4%) | 5 (1.5%) | |
| **PCA-related factor** | | | | |
| Background dose of fentanyl in PCA (μg/kg/h) | 0.356 (0.315–0.407) | 0.353 (0.313–0.404) | 0.370 (0.328–0.424) | < 0.001 |
| Adjuvant nefopam in PCA | 1,909 (88.8%) | 1,606 (88.6%) | 303 (89.9%) | 0.55 |
| Antiemetic, Ramosetron | 2,077 (96.6%) | 1,752 (96.7%) | 325 (96.4%) | 0.94 |

BMI, body mass index; ASA, American Society of Anesthesiologists; TIVA, total intravenous anesthesia; PCA, patient-controlled analgesia

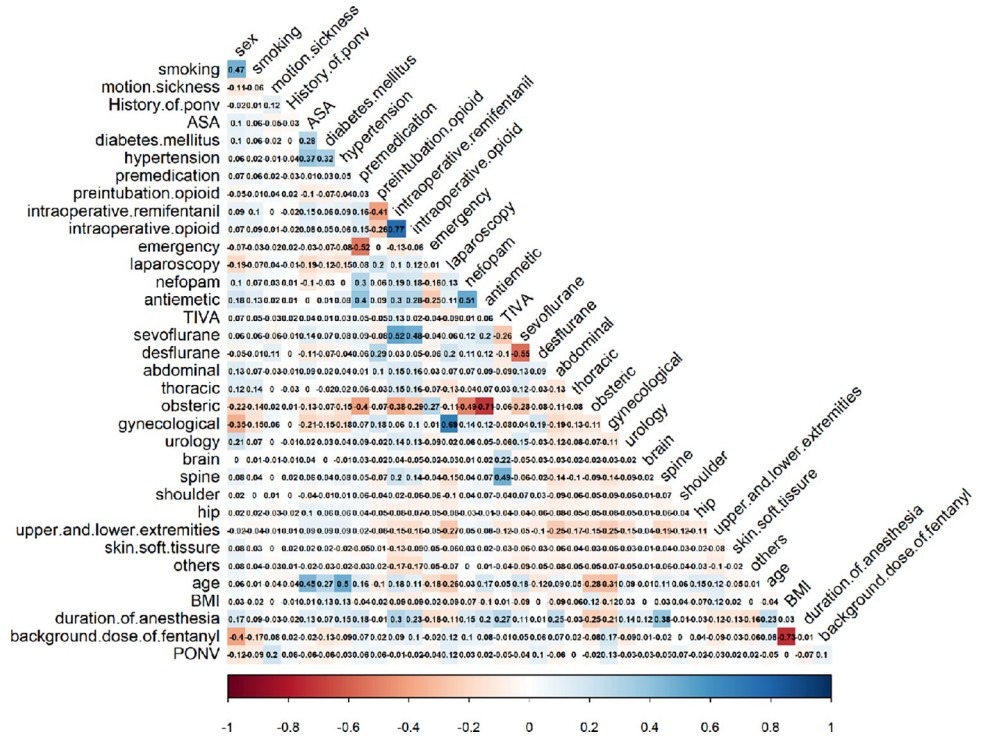

**Fig 1. Correlation between variables.**

decision tree was 0.561 (95% CI, 0.498–0.625), random forest was 0.610 (95% CI, 0.552–0.668), gradient boosting machine was 0.580 (95% CI, 0.520–0.639), support vector machine (SVM) was 0.649 (95% CI, 0.592–0.707), artificial neural network (ANN) was 0.686 (95% CI,

| | | |
|---|---|---|
| **Age** | Emergency | |
| **Sex (female)** | **Laparoscopic surgery** | |
| **BMI** | **Type of surgery** | Age |
| History of smoking | Abdominal | Sex (female) |
| **History of motion sickness** | Thoracic | BMI |
| **History of PONV** | Obstetric | History of motion sickness |
| **ASA physical status** | Gynecological | History of PONV |
| Diabetes mellitus | Urology | ASA physical status |
| **Hypertension** | Brain | Hypertension |
| **Duration of anesthesia** | Spine | Duration of anesthesia |
| **Premedication (anticholinergic)** | **Shoulder** | Premedication (anticholinergic) |
| **Preintubation opioid** | Hip | Preintubation opioid |
| Anesthetics in G/A | Upper and lower extremities | Laparoscopic surgery |
| Sevoflurane | Skin, soft tissue | Type of surgery |
| Desflurane | Others | Shoulder |
| TIVA | **Background dose of fentanyl in PCA** | Background dose of fentanyl in PCA |
| Intraoperative remifentanil infusion | Adjuvant Nefopam in PCA | |
| Intraoperative opioids (fentanyl, meperidine) | Antiemetic (ramosetron) | |

Recursive feature elimination →

**Fig 2. Feature selection process by recursive feature elimination on the training dataset.**

**Table 3. Performance of the machine learning and Apfel model.**

| Model | AUROC (95% CI) | Accuracy (95% CI) | Sensitivity (95% CI) | Specificity (95% CI) |
|---|---|---|---|---|
| LR | 0.576 | 0.544 | 0.88 | 0.20 |
|  | (0.520–0.633) | (0.504–0.582) | (0.84–0.91) | (0.15–0.24) |
| KNN | 0.597 | 0.649 | 0.87 | 0.20 |
|  | (0.537–0.656) | (0.611–0.686) | (0.83–0.90) | (0.15–0.26) |
| DT | 0.561 | 0.550 | 0.86 | 0.17 |
|  | (0.498–0.625) | (0.510–0.589) | (0.82–0.89) | (0.13–0.22) |
| RF | 0.610 | 0.652 | 0.88 | 0.22 |
|  | (0.552–0.668) | (0.614–0.689) | (0.84–0.91) | (0.17–0.28) |
| GBM | 0.580 | 0.602 | 0.86 | 0.19 |
|  | (0.520–0.639) | (0.564–0.640) | (0.83–0.90) | (0.14–0.24) |
| SVM | 0.649 | 0.717 | 0.88 | 0.25 |
|  | (0.592–0.707) | (0.681–0.752) | (0.84–0.90) | (0.19–0.33) |
| **ANN** | **0.686** | 0.593 | 0.92 | 0.24 |
|  | **(0.630–0.742)** | (0.554–0.631) | (0.89–0.95) | (0.20–0.29) |
| Apfel | 0.643 | 0.523 | 0.92 | 0.21 |
|  | (0.596–0.690) | (0.484–0.562) | (0.88–0.95) | (0.17–0.26) |

AUROC, area under the receiver operating characteristic; LR, logistic regression; KNN, k-nearest neighbors; DT decision tree; RF, random forest; GBM, gradient boosting machine; SVM, support vector machine; ANN, artificial neural networks

0.630–0.742), and Apfel score was 0.643 (95% CI, 0.596–0.690). The ANN showed the largest AUROC (0.686, 95% CI, 0.630–0.742), as shown in Fig 3. The SVM showed the highest accuracy (0.717, 95% CI, 0.681–0.752).

The entire code used in this study is available online without restrictions (https://github.com/jgshim/IV-PCA). The detailed hyperparameters of the machine learning model used in this study can be found in the S1 Table.

## Discussion

We analyzed and compared the predictive ability of seven machine learning approaches and the Apfel model to predict PONV during 24 h after surgery. The results showed that the ANN method had the largest AUROC for identifying PONV using clinical data. The key findings were as follows: (1) machine learning models such as ANN and SVM showed better performance than the Apfel model and (2) feature selection using recursive feature elimination improved human insight into complex and non-linear models associated with PONV. To our knowledge, this is the first study to predict the occurrence of PONV by comparing various classification machine learning approaches with the Apfel model.

Conventional machine learning approaches generally work efficiently with traditional datasets and allow for nonlinear relationships between predictors but may deteriorate with high-dimensional problems [24]. We explored the number of selected features using the wrapped algorithm used in the recursive feature elimination procedure. By means of dimensionality reduction, the dependencies and collinearity that may exist in the model can be eliminated to improve performance.

Although volatile anesthetics was an important factor of PONV in a previous study [25], in this study, the type of volatile anesthetics or intravenous anesthesia was not helpful in predicting PONV. An increase in the duration of anesthesia was associated with a reduction in

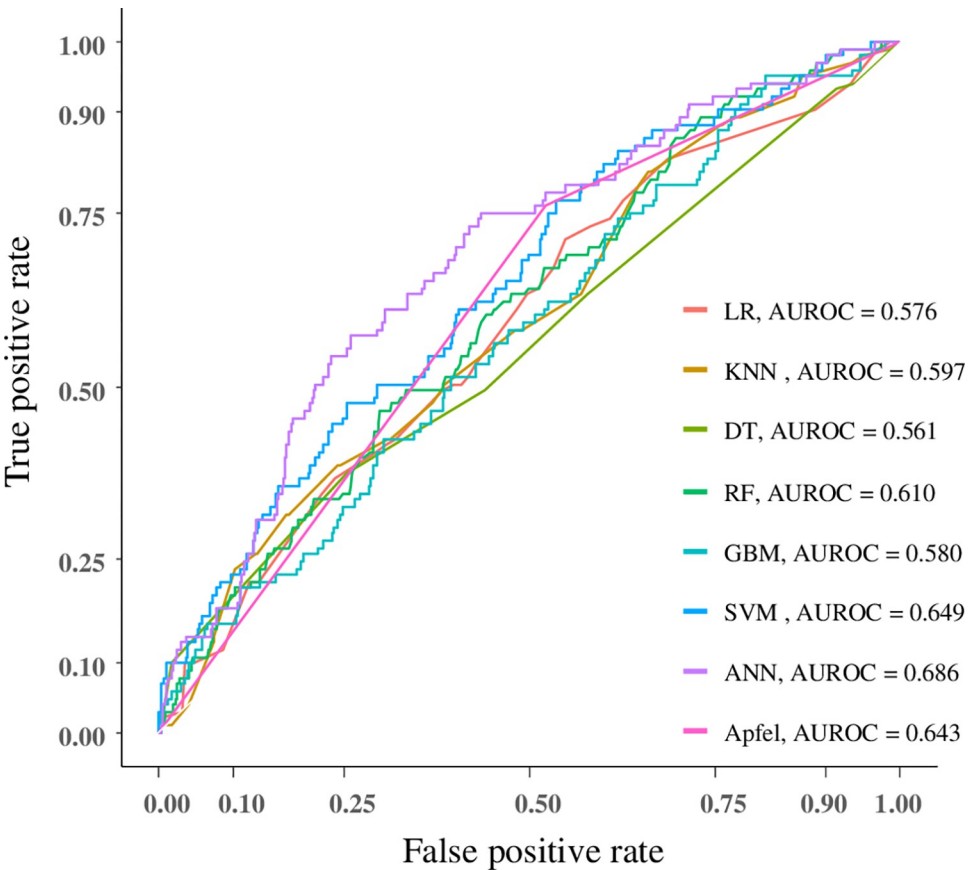

**Fig 3. Areas under the receiver operating curve for the machine learning and Apfel model.**

PONV, which is inconsistent with previous results [26]. The increase in PONV with the use of preintubation opioids or laparoscopic surgery is evidenced in previous studies [27, 28]. Hypotension occurring during shoulder surgery may be a major factor of PONV and was used as one of the features in our study [12, 29].

We believe that anesthesiologists in the operating room can help manage PONV. For example, considering the possibility of PONV, the ANN or SVM model could be useful in deciding whether to take preemptive measures, such as preparing an antiemetic in advance or continuing follow-up and observation. Furthermore, cost-effective management will be possible because the models require only 13 clinical variables to identify patients at a high risk of PONV.

The patients with chemotherapy history are at high risk for opioid induced PONV [30]. However, the number of cases of postchemotherapy patients were very small. Even if they had a history of cancer, it was not clear whether they had received chemotherapy. Thus, we were not able to include the postchemotherapy patient group as the input variable. Including chemotherapy as an input factor in further studies may improve the performance of the PONV prediction model.

It is clear that PONV is a distressing side effect in patients. A suitable screening test for PONV should include adequate sensitivity and specificity, and be acceptable to both patient and medical practitioners. Having high sensitivity but low specificity may lead to inappropriate preemptive measures. For instance, a patient with a low risk of PONV might be given antiemetics. Therefore, careful attention should be paid when used as a screening tool.

There are some limitations to our study. First, we are not sure that the amount of data we used was enough to work on machine-learning problems, considering the complexity of the problem and nonlinear algorithms. Further data about PONV should be collected to improve the predictive power. Second, one of the PCA teams in the anesthesiology department visited only once during the day after surgery and asked about the effects and complications of PCA. As a result, because of recall bias, the PONV occurrence rate may have been underestimated. Third, because our study analyzed data from a single center, it might not be possible to apply our model to a wider population. Further studies are needed, with large heterogeneous samples, to improve generalizability.

## Conclusions

In summary, we developed and compared various machine learning models and the Apfel model to predict the occurrence of PONV using IV-PCA. We expect our results to help reduce PONV by helping clinicians predict it and take preemptive actions.

## Supporting information

**S1 Table. Optimal hyperparameters of all machine learning models.**
(DOCX)

## Author Contributions

**Conceptualization:** Jae-Geum Shim, Kyoung-Ho Ryu, Eun-Ah Cho, Sung Hyun Lee.

**Data curation:** Kyoung-Ho Ryu, Eun-Ah Cho, Jin Hee Ahn, Yun Byeong Cha, Goeun Lim, Sung Hyun Lee.

**Formal analysis:** Jae-Geum Shim, Kyoung-Ho Ryu, Jin Hee Ahn.

**Methodology:** Jae-Geum Shim.

**Project administration:** Eun-Ah Cho.

**Resources:** Goeun Lim.

**Supervision:** Jin Hee Ahn, Sung Hyun Lee.

**Writing – original draft:** Jae-Geum Shim.

**Writing – review & editing:** Sung Hyun Lee.

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
