## [Decision Letter · Decision Letter 0]

11 Sep 2022

PONE-D-22-18403Machine learning for prediction of postoperative nausea and vomiting in patients with intravenous patient-controlled analgesiaPLOS ONE

Dear Dr. Lee,

Thank you for submitting your manuscript to PLOS ONE. After careful consideration, we feel that it has merit but does not fully meet PLOS ONE’s publication criteria as it currently stands. Therefore, we invite you to submit a revised version of the manuscript that addresses the points raised during the review process.

We look forward to receiving your revised manuscript.

Kind regards,

Jianhong Zhou

Staff Editor

PLOS ONE

Journal Requirements:

Additional Editor Comments:

Specifically, PLOS ONE requires that all experiments, statistics and other analyses are performed to a high technical standard, described in sufficient detail and adhere to appropriate reporting guidelines and community standards. After review of your manuscript, we had several concerns with the analyses, conclusions and quality of the scientific reporting. Please provide the following information:

-Why Apfel was chosen as the comparison model? Are there any other existing approaches?

-Data collection rationale and criteria, e.g. why pick the time range July 2019 and July 2020? Was sample size calculated before data collection to determine if the final data size is appropriate in light of the study design and your conclusion?

-Feature selection strategies, i.e. how were those 21 and 34 variables identified? For the 13 features that were used for training, how did they correlate with each other?

-Definitions of the model performance measurements such as accuracy, sensitivity and specificity

- Machine learning model details, i.e. any specific parameters used or they are default in the R packages? How are the data normalized?

In addition, PLOS journals require authors to make all data underlying the findings described in their manuscript fully available without restriction at the time of publication (https://journals.plos.org/plosone/s/data-availability>). This policy is aimed to ensure that other researchers can reproduce the analysis. We understand that there are privacy and ethical restrictions of your raw data underlying the prediction, if possible,  please deposit your de-identified data to a public data repository or include it in the Supporting Information files and update your data availability statement accordingly. 

Minor comments:

-In Table 2, please highlight the models that performed the best for each measurement

-Please provide figure legends

-Please discuss the likely underlying reasons for why ANN and SVM were better than other models

Finally, please present the conclusions that are supported by your results. Given that the accuracy and specificity of the models are not that high, we suggest modifying your conclusions to avoid overstating the implications.

Reviewers' comments:

Reviewer's Responses to Questions

**Comments to the Author**

1. Is the manuscript technically sound, and do the data support the conclusions?

Reviewer #1: Yes

2. Has the statistical analysis been performed appropriately and rigorously? 

Reviewer #1: Yes

3. Have the authors made all data underlying the findings in their manuscript fully available?

Reviewer #1: Yes

4. Is the manuscript presented in an intelligible fashion and written in standard English?

Reviewer #1: Yes

5. Review Comments to the Author

Reviewer #1: Dear Author.

It is a well conducted study and nicely written manuscript.

My points are:

1. Did you take the postoperative pain into consideration while applying machine learning for PONV? The postoperative pain is also a major cause of PONV.

2. Opioid induced PONV (PCA based) in patients with high risk of PONV like post chemotherapy patients or previous history of PONV? This should be considered also during the ML data processing or should include in discussion and limitations.

Comparison of palonosetron and dexamethasone with ondansetron and dexamethasone for postoperative nausea and vomiting in postchemotherapy ovarian cancer surgeries requiring opioid-based patient-controlled analgesia: A randomised, double-blind, active controlled study. Indian J Anaesth. 2018 Oct;62(10):773-779. doi: 10.4103/ija.IJA_437_18. PMID: 30443060; PMCID: PMC6190431.

6. PLOS authors have the option to publish the peer review history of their article (what does this mean?). If published, this will include your full peer review and any attached files.

Reviewer #1: No

---

## [Author Response · Author response to Decision Letter 0]

26 Sep 2022

Response to Reviewers

Academic editor

1. Why Apfel was chosen as the comparison model? Are there any other existing approaches?

Response: Apfel’s simplified risk score is the most widely used tool for risk stratification of postoperative nausea and vomiting (PONV). It needs only 4 independent risk factors including female sex, nonsmoking status, history of PONV or motion sickness, and postoperative opioid use. Therefore, we chose Apfel as the comparison model. However, the Apfel model does not guarantee an accurate prediction for the risk of PONV.[1, 2] Recent studies using various logistic regression or machine learning models have shown poor results.[3-5] Also, there was a study that used the Apfel model as a comparison model in predicting PONV.[1]

References

1. Chae D, Kim SY, Song Y, Baek W, Shin H, Park K, et al. Dynamic predictive model for postoperative nausea and vomiting for intravenous fentanyl patient-controlled analgesia. Anaesthesia. 2020;75(2):218-26. Epub 2019/09/19. doi: 10.1111/anae.14849. PubMed PMID: 31531854.

2. van den Bosch JE, Kalkman CJ, Vergouwe Y, Van Klei WA, Bonsel GJ, Grobbee DE, et al. Assessing the applicability of scoring systems for predicting postoperative nausea and vomiting. Anaesthesia. 2005;60(4):323-31. Epub 2005/03/16. doi: 10.1111/j.1365-2044.2005.04121.x. PubMed PMID: 15766334.

3. Hu YJ, Ku TH. Pattern discovery from patient controlled analgesia demand behavior. Comput Biol Med. 2012;42(10):1005-11. Epub 2012/09/11. doi: 10.1016/j.compbiomed.2012.08.002. PubMed PMID: 22959278.

4. Palazzo M, Evans R. Logistic regression analysis of fixed patient factors for postoperative sickness: a model for risk assessment. Br J Anaesth. 1993;70(2):135-40. Epub 1993/02/01. doi: 10.1093/bja/70.2.135. PubMed PMID: 8435254.

5. Yuh-Jyh H, Jia-Ying S, Tien-Hsiung K. Predicting Postoperative Nausea and Vomiting Under Patient-Controlled Analgesia Medication: A Study of Machine Learning Approaches. Primary Health Care: Open Access. 2017;7(3):1-6.

2. Data collection rationale and criteria, e.g. why pick the time range July 2019 and July 2020? Was sample size calculated before data collection to determine if the final data size is appropriate in light of the study design and your conclusion?

Response: In this study, the sample size of machine learning models was not determined prior to the data collection. The PCA regimen frequently changed in the hospital where the author works. The time between July 2019 and July 2020 was selected as the data collection period as the fentanyl-based IV-PCA regimen remained constant. Unfortunately, an accurate calculation method for machine learning sample size is lacking at this time. Therefore, considering the rule of thumb, we needed at least 1,000 samples per class. The sample size of the non-PONV group was suitable in this study, but the number of samples of PONV group was insufficient. As a result, we oversampled the minority class and balanced the low incidence of PONV in this study.

3. Feature selection strategies, i.e. how were those 21 and 34 variables identified? For the 13 features that were used for training, how did they correlate with each other?

Response: First, 21 variables were selected based on the previous literature and our clinical experience. Then categorical variables with n levels were transformed into n variables, each with two levels. As a result, 34 variables were considered as input variables for the model. Then, 13 variables with the best prediction performance were selected by the Recursive Feature Elimination (RFE) algorithm. The correlation of 13 variables that were used for the training is shown in the figure below. There was a weak association between PONV and previous motion sickness. We also found a very weak association between PONV and female, laparoscopy surgery, and background dose of fentanyl.

4. Definitions of the model performance measurements such as accuracy, sensitivity and specificity

Response: Thank you for suggesting important points. We have added a table of confusion matrix and revised manuscript.

“The confusion matrix is used for summarizing the performance of a classification problem as shown in Table x. Accuracy, sensitivity, and specificity are described in terms of true positive (TP), true negative (TN), false negative (FN) and false positive (FP).

The accuracy of model is the ratio of correct predictions to total predictions made and is defined as

Accuracy = (TN + TP) / (TN+TP+FN+FP)

The sensitivity of model is the proportion of actual positive cases that are correctly identified and is defined as

Sensitivity = TP / (TP + FN)

The specificity of model is the proportion of actual negative cases that are correctly identified and is defined as

Specificity = TN / (TN + FP)”

Table 1. Confusion matrix

 Actual

 Positive Negative

Predicted Positive TP TN

 Negative FP FN

TP, true positive; TN, true negative; FN, false negative; FP, false positive

5. Machine learning model details, i.e. any specific parameters used or they are default in the R packages? How are the data normalized?

Response: The S1 Table contains the details of the hyperparameters that we used for our machine learning model. In our study, continuous variables were transformed into values between 0 and 1 by minimum-maximum normalization, implemented in the caret package in the R software. We added the S1 Table in manuscript.

“The detailed hyperparameters of the machine learning model used in this study can be found in the S1 table.”

S1 Table. Optimal hyperparameters of all machine learning models

Model Optimal hyperparameters

LR nIter* = 31

KNN k = 8

DT Maximum depth = 5

Criterion = Gini index

RF mtry** = 3

GBM Maximum depth = 3

Number of estimators = 50,

Gamma = 0

SVM degree = 3, scale = 0.1 and C = 1.0

ANN Number of hidden layers = 2

Number of nodes in a layer = 32, 16

LR, logistic regression; KNN, k-nearest neighbors; DT, decision tree; RF, random forest; GBM, gradient boosting machine; SVM, support vector machine; ANN, artificial neural networks

*nIter indicates an integer, describing the number of iterations for which boosting should be run.

**mtry indicates the number of variables available for splitting at each tree node.

6. In addition, PLOS journals require authors to make all data underlying the findings described in their manuscript fully available without restriction at the time of publication (https://journals.plos.org/plosone/s/data-availability>). This policy is aimed to ensure that other researchers can reproduce the analysis. We understand that there are privacy and ethical restrictions of your raw data underlying the prediction, if possible, please deposit your de-identified data to a public data repository or include it in the Supporting Information files and update your data availability statement accordingly.

Response: We understand the guidelines of PLOS journals. There were ethical restrictions of raw data, and thus we deposited de-identified and preprocessed data to a public data repository. (https://github.com/jgshim/IV-PCA)

7. Minor comments:

In Table 2, please highlight the models that performed the best for each measurement

Response: We highlighted the models in Table 2.

Please provide figure legends

Response: We provided figure legends.

Please discuss the likely underlying reasons for why ANN and SVM were better than other models

Response: Our primary analysis goal was to compare the prediction ability of machine learning approaches to Apfel model in terms of the area under the receiver-operating characteristics curve (AUROC). As shown in previous literature, the analysis of AUROC is a valuable tool to evaluate predictive models.[6] Table x shows the comparison of test AUROC to predict PONV according to the model. The AUROC of ANN and SVM were larger and therefore better than all other models.

Reference

6. Zou KH, O'Malley AJ, Mauri L. Receiver-operating characteristic analysis for evaluating diagnostic tests and predictive models. Circulation. 2007;115(5):654-7. Epub 2007/02/07. doi: 10.1161/circulationaha.105.594929. PubMed PMID: 17283280.

8. Finally, please present the conclusions that are supported by your results. Given that the accuracy and specificity of the models are not that high, we suggest modifying your conclusions to avoid overstating the implications.

Response: We fully agree with your opinion. We revised our manuscript.

“It is clear that PONV is a distressing side effect in patients. A suitable screening test for PONV should include adequate sensitivity and specificity, and be acceptable to both patient and medical practitioners. Having high sensitivity but low specificity may lead to inappropriate preemptive measures. For instance, a patient with a low risk of PONV might be given anti-emetics. Therefore, careful attention should be paid when used as a screening tool. 

Response to peer reviewers

Reviewer #1:

1. Did you take the postoperative pain into consideration while applying machine learning for PONV? The postoperative pain is also a major cause of PONV.

Response:

We fully agree with your opinion. The postoperative pain is a major cause of PONV. Unfortunately, postoperative pain was not used as an input variable in our predictive model for the following two reasons.

Initially the postoperative pain score was obtained from EMR in raw data. However, we concluded that the postoperative pain score recorded was inconsistent due to the following reasons. Firstly, the frequency and dosage of the rescue analgesia were inconsistent. Secondly, the pain scores fluctuated depending on the time of the day, situation (i.e. ambulation, during sleep), and environment.

2. Opioid induced PONV (PCA based) in patients with high risk of PONV like post chemotherapy patients or previous history of PONV? This should be considered also during the ML data processing or should include in discussion and limitations.

Comparison of palonosetron and dexamethasone with ondansetron and dexamethasone for postoperative nausea and vomiting in postchemotherapy ovarian cancer surgeries requiring opioid-based patient-controlled analgesia: A randomised, double-blind, active controlled study. Indian J Anaesth. 2018 Oct;62(10):773-779. doi: 10.4103/ija.IJA_437_18. PMID: 30443060; PMCID: PMC6190431

Response:

As the reviewer pointed out, postchemotherapy patients or those with a previous history of PONV are at high risk for PONV. The previous history of PONV was used as an input to our model. However, the number of cases of postchemotherapy patients were very small. Even if they had a history of cancer, it was not clear whether they had received chemotherapy. Thus, we were not able to include the postchemotherapy patient group as the input variable. Including chemotherapy as an input factor in further studies may improve the performance of the PONV prediction model.

We revised manuscript in “Discussion” section.

“The patients with chemotherapy history are at high risk for opioid induced PONV.[7] However, the number of cases of postchemotherapy patients were very small. Even if they had a history of cancer, it was not clear whether they had received chemotherapy. Thus, we were not able to include the postchemotherapy patient group as the input variable. Including chemotherapy as an input factor in further studies may improve the performance of the PONV prediction model.

Reference

7. Kumar A, Solanki SL, Gangakhedkar GR, Shylasree TS, Sharma KS. Comparison of palonosetron and dexamethasone with ondansetron and dexamethasone for postoperative nausea and vomiting in postchemotherapy ovarian cancer surgeries requiring opioid-based patient-controlled analgesia: A randomised, double-blind, active controlled study. Indian J Anaesth. 2018;62(10):773-9. Epub 2018/11/18. doi: 10.4103/ija.IJA_437_18. PubMed PMID: 30443060; PubMed Central PMCID: PMCPMC6190431.

---

## [Decision Letter · Decision Letter 1]

8 Nov 2022

Machine learning for prediction of postoperative nausea and vomiting in patients with intravenous patient-controlled analgesia

PONE-D-22-18403R1

Dear Dr. Lee,

We’re pleased to inform you that your manuscript has been judged scientifically suitable for publication and will be formally accepted for publication once it meets all outstanding technical requirements.

Kind regards,

Sathishkumar V E

Academic Editor

PLOS ONE

Additional Editor Comments (optional):

Reviewers' comments:

Reviewer's Responses to Questions

**Comments to the Author**

1. If the authors have adequately addressed your comments raised in a previous round of review and you feel that this manuscript is now acceptable for publication, you may indicate that here to bypass the “Comments to the Author” section, enter your conflict of interest statement in the “Confidential to Editor” section, and submit your "Accept" recommendation.

Reviewer #1: All comments have been addressed

2. Is the manuscript technically sound, and do the data support the conclusions?

Reviewer #1: Yes

3. Has the statistical analysis been performed appropriately and rigorously? 

Reviewer #1: Yes

4. Have the authors made all data underlying the findings in their manuscript fully available?

Reviewer #1: Yes

5. Is the manuscript presented in an intelligible fashion and written in standard English?

Reviewer #1: Yes

6. Review Comments to the Author

Reviewer #1: Dear Authors, Thank you for the revision. All the comment has been addressed by the authors. The article is good to go for the publications

7. PLOS authors have the option to publish the peer review history of their article (what does this mean?). If published, this will include your full peer review and any attached files.

Reviewer #1: **Yes: **Sohan Lal Solanki

---

## [Editor Report · Acceptance letter]

14 Dec 2022

PONE-D-22-18403R1 

Machine learning for prediction of postoperative nausea and vomiting in patients with intravenous patient-controlled analgesia 

Dear Dr. Lee:

I'm pleased to inform you that your manuscript has been deemed suitable for publication in PLOS ONE. Congratulations! Your manuscript is now with our production department. 

Kind regards, 

on behalf of

Dr. Sathishkumar V E 

Academic Editor

PLOS ONE